# Novel Synthesis of Choline-Based Amino Acid Ionic Liquids and Their Applications for Separating Asphalt from Carbonate Rocks

**DOI:** 10.3390/nano9040504

**Published:** 2019-04-01

**Authors:** Zisheng Zhang, Ning Kang, Jingjing Zhou, Xingang Li, Lin He, Hong Sui

**Affiliations:** 1School of Chemical Engineering and Technology, Tianjin University, Tianjin 300072, China; zzhang@uottawa.ca (Z.Z.); ningkang@tju.edu.cn (N.K.); zhoujj.zm@gmail.com (J.Z.); lxg@tju.edu.cn (X.L.); 2National Engineering Research Center of Distillation Technology, Tianjin 300072, China; 3Department of Chemical & Biological Engineering, University of Ottawa, 161 Louis Pasteur St., Ottawa, ON K1N6N5, Canada; 4Collaborative Innovation Center of Chemical Science and Engineering, Tianjin 300072, China

**Keywords:** amino acid ionic liquid, carbonate asphalt rock, COSMOtherm, molecular dynamics simulation

## Abstract

In this study, a series of choline-based amino acid ionic liquids have been synthesized in an economic method and are used to assist solvents with extracting asphalt from carbonate rocks. All of the ionic liquids perform well in extracting asphalt, especially choline histidine, by which the single-step recovery of asphalt is up to 91%. Furthermore, oil product with higher quality (fewer solids entrained) is also obtained. Molecular dynamics simulation and thermodynamic equilibrium method are applied to investigate the role of amino acid ionic liquids via interaction energy calculation and surface free energy calculations. The simulation results suggest that the ionic liquid phase is beneficial for the transfer of oil fraction from the carbonate surface to the organic solvent phase. Moreover, the results of simulated calculation show that the introduction of a functional group with conjugated structures into ionic liquid, such as an imidazole ring and a benzene ring, is beneficial for enhancing oil recovery, which are in accordance with the results of experimental tests.

## 1. Introduction

Unconventional oils, with geological reserves of up to about two-thirds of total petroleum, have been considered as a major part of petroleum energy [1]. The Indonesian asphalt rocks, one kind of oil sands with bitumen content up to 50 wt.%, was reported to possess over 3 billion tons of bitumen geological in situ reserves. Due to the huge reserves, it has been attracting increasing attention [2,3]. Different from the Canadian oil sands, the interaction between the Indonesian asphalt and the host rocks is much stronger than that between Canadian bitumen and sand grains. It is proved that the traditional water-flooding process works poorly in separating asphalt from carbonate rocks [4]. To unlock this kind of petroleum ore, other different methods have been proposed, such as solvent extraction [1,5,6,7], ionic liquid-assisted solvent extraction [8,9,10,11], and pyrolysis [12,13,14,15]. 

Among these methods, ionic liquid-assisted solvent extraction seems to be a potential one. Compared with a solvent extraction method, it not only increases the recovery of bitumen, but also decreases the solids entrained in bitumen [8,9]. However, ionic liquids (ILs) that consist of imidazolium or pyridinium cations and halide anions are not only toxic to various degrees but also expensive [16,17]. In addition, the synthetic process of some ILs is complex or even not “green” as well. How to design a cheap, low toxic, and efficient IL with an environmentally friendly synthetic process, is an important research area related to the IL-assisted solvent extraction.

During the past decades, a kind of ‘‘fully green’’ ionic liquid with amino acids as anions or cations have been obtained [18,19,20,21]. The so-called amino acid ionic liquids (AAILs) possess good biodegradability, low toxicity and relatively low cost. Take choline-based AAILs as an example, they are synthesized by choline hydroxide and amino acid [22]. They are well known to be nontoxic and biodegradable, and have been applied well in the pre-treatment of biomass. However, one of raw materials of choline-based AAILs, choline hydroxide is unstable and expensive. Accordingly, the reduction of the synthetic cost of choline-based AAILs becomes one of the main challenges before the potential application of choline-based AAILs.

On the other hand, because of the complexity of unconventional oil ores, the roles of ILs in separating asphalt from carbonate rocks are still unclear. Some primary attempts have been made to reveal the mechanisms of IL-assisted solvent extraction for the oil-solid separation. It is reported that the ILs could modify the heavy oil surface properties through surface adsorption [23]. Further tests by Sui et al. using silica sands indicated the presence of ILs could reduce the adhesion force between bitumen and quartz sand surface [24]. With the assistance of advanced computation technology, the molecular behaviors of ILs during the solvent extraction of Athabasca oil sands have been investigated by Li et al. [11,25,26]. The simulation results show that ILs would perform as a “carrier” between silica surface and solvent-diluted bitumen. These previous works provide useful information for further investigation of IL-assisted processes of separating asphalt from carbonate rocks.

Accordingly, the objectives of this study are to: (i) synthesize a series of choline-based AAILs in an economic way; (ii) test their feasibility in extracting asphalt from carbonate rock surface; (iii) obtain the optimal AAIL for enhancing oil recovery; (iv) understand its roles in separating asphalt oil from carbonate rocks; and (v) reveal the plausible mechanisms of AAIL-enhanced oil recovery.

## 2. Experimental

### 2.1. Materials

Chemical reagents, including glycine (99 wt.%), serine (99 wt.%), proline (99 wt.%), histidine (99 wt.%), phenylalanine (99 wt.%), potassium hydroxide (96 wt.%), choline chloride (99 wt.%), absolute ethanol, acetonitrile, and toluene, were purchased and used at their analytical grade from Aladdin Bio-Chem Technology, Shanghai, China. The physicochemical properties of amino acids were shown in Table 1. The Indonesia asphalt carbonate rocks used in this work were from the Buton Island in Indonesia. The asphalt rock samples were analysed using the Dean-Stark standard method [27] and American Society for Testing Material Standard D4124 [3]. The composition is determined to be: 30.06 wt.% asphalt, 0.17 wt.% water, and 69.77 wt.% solids. Furthermore, the composition of asphalt is divided into four fractions (SARA): saturate (15.97 wt.%), aromatic (19.58 wt.%), resin (38.76 wt.%), and asphaltene (25.69 wt.%). The size of solids in asphalt rocks ranges from 10 to 200 μm and d_50_ is 29 μm. The major component of solids is calcium carbonate. Detailed mineral analysis has reported in previous work [4].

### 2.2. Synthesis and Characterization of AAILs

In this study, a new method, one-pot synthesis, was proposed to synthesize the AAILs, including choline glycine (ChGly), choline histidine (ChHis),choline serine (ChSer), choline proline (ChPro), and choline phenylalanine (ChPhe). The choline chloride (ChCl) and amino acids were added into potassium hydroxide-ethanol solution which was precalibrated (Equation (1)). The mole ratio of choline chloride, potassium hydroxide and amino acid is 1:1:1.05. The reaction lasted for 8 h at 30 °C under ambient pressure, followed by filtering to separate the nonsoluble by-product potassium chloride from reaction mixture. Then the liquid was evaporated to obtain the crude product. The crude product was washed by acetonitrile eluent, and then the excess unreacted amino acids were precipitated out. Then, the AAILs were obtained, shown in Figure 1.
(1)$$R\;-\;{\mathrm{CHNH}}_{2}\;\mathrm{COOH}\;+\;\mathrm{KOH}\;+\;\mathrm{ChCl}\;\overset{\mathrm{EtOH}}{\to }\;R\;-\;{\mathrm{CHNH}}_{2}\;\mathrm{COOCh}\;+\;\mathrm{KCl}\;+\;H_{2}O$$

To confirm the successful synthesis of the designed materials, the structures of the AAILs were characterized by nuclear magnetic resonance (NMR) (VARIAN, INOVA 500 MHz, Palo Alto, CA, USA), fourier transform infrared spectroscopy (FT-IR) (Bio-Rad, FTS6000, Cambridge, MA, USA), differential scanning calorimeter (DSC) (NETZSCH, DSC204, Selb, Germany) and thermogravimetric analysis (TGA) (NETZSCH, TG209, Selb, Germany). During the NMR test, samples were prepared with deuteroxide (D_2_O) as the solvent. The ^1^H NMR was operated at proton frequency of 500 MHz at 30 °C. The FT-IR detection was conducted at a wavenumber resolution of 4 cm^−1^ by 100 scans at ambient conditions. The glass transition temperatures (T_g_) of the AAILs were obtained by DSC. The samples were prepared in hermetically sealed Al pans and cooled to −100 °C, followed by heating to 30 °C at the rate of 5 °C/min. The decomposition temperatures (T_d_) of the AAILs were determined by TGA. The samples were heated at a rate of 10 °C/min from 20 °C to 800 °C under nitrogen atmosphere.

### 2.3. Solvent Extraction Assisted by AAILs

The AAILs were used to work together with toluene to enhance asphalt recovery from carbonate asphalt rocks. Herein, the AAIL (5.00 ± 0.01 g) was mixed with toluene at the weight ratio of 1:2. Then, the ore sample (5.00 ± 0.01 g) was added into the liquid mixture for 30 min agitation at 30 °C. After the agitation, the slurry was transferred into a centrifuge tube for centrifugation at 7000 rpm for 10 min. The slurry was divided into three phases: the upper asphalt toluene solution, the middle AAIL phase and carbonate rocks in the bottom. The supernatant was removed for distillation to obtain the extracted asphalt. The recovery of asphalt was calculated by Equation (2):
(2)$$R=\frac{m_{b}/m_{o}}{R_{o}}\times 100\%$$
where *m_b_* presents the mass of the extracted heavy hydrocarbons (g), *m_o_* is the mass of the ore sample (g) and *R_o_* is the heavy oil content in ore sample.

To confirm if any AAIL was dissolved in the oil phase, FT-IR was applied to analyse the extracted asphalt. The solids entrainment in the asphalt was tested by Equation (3) accordingly.
*η* = *m_s_*/*V_a_*(3)
where *m_s_* presents the mass of the solids entrainment in asphalt-toluene solution (mg) and *V_a_* is the volume of asphalt-toluene solution (mL). The procedure details are given elsewhere [4].

For comparison, pure solvent extraction (5.00 g of asphalt carbonate rocks was washed by 10.00 g of toluene only) was conducted using the same procedure. Each test was repeated at least three times until the credible results were obtained.

### 2.4. Interfacial Tension Measurements

To investigate the role of AAILs in enhancing oil recovery from carbonate asphalt rock, surface tension and interfacial tension were measured by dynamic contact angle tensiometer (pendant drop method, SL200B, Kono, Seattle, WA, USA). The surface tension of AAILs and interfacial tension of AAIL-organic solvent (i.e., toluene) interface were measured at 30 °C.

### 2.5. Computational Simulation

To deeply understand the roles of ILs, the molecular dynamics simulation and thermodynamic equilibrium method were employed for energy calculation using Material Studio software (7.0 package, Accelrys, San Diego, CA, USA) and COSMOtherm software (C30-1301, COMSOlogic, Leverkusen, Germany), respectively. All the ions of IL, solvent, bitumen component (SARA) and mineral surface models were drawn by the Material Studio software. Particularly, calcite crystal, the main component of carbonate asphalt rock [4], was selected to represent the mineral and four typical molecular structures were selected to denote the SARA fractions for simplification, as shown in Figure 2 [11,28,29,30,31].

The surface free energies of compounds at the interface among the three phases of extraction system in Section 2.3 are used to investigate the roles of AAIL in enhancing oil recovery from carbonate asphalt rock. In this study, surface free energies of SARA at AAIL-asphalt (or AAIL-asphalt toluene solution) interfaces were calculated by COSMOtherm software based on the COSMO-RS theory at 30 °C under ambient pressure. The total free energy of the solute at the interface can be used as a significant and thermodynamically rooted descriptor for the determination of surface activity in a solution [11]. The calculations of free energy at 30 °C under ambient pressure were performed for SARA at the SARA fraction-IL interface and toluene-IL interface.

In addition, the interaction energies between AAILs (or asphalt) and carbonate rocks were simulated by Material Studio software using focite module with COMPASS force field in canonical ensemble (NVT) for the established adsorption system at 30°C and a simulation time of 50 ps. The temperature was controlled by a Berendsen thermostat. In present work, the valence terms (bond energy, angle energy, torsion energy, out-of-plane energy, and cross-coupling energy) were ignored and only the non-bond interactions (van der Waals energy and Columbic interaction) were used in calculating the interaction energies. The interaction energies were calculated by Equation (4):
E_interaction_ = E_total_ − (E_adsorbate_ + E_surface_)(4)
where E_total_, E_surface_ and E_adsorbate_ represent energies of the adsorption system, carbonate rocks surface, and adsorbate IL (or asphalt), respectively. The more negative the interaction energy is, the stronger the adsorption is. The detailed procedures can be found elsewhere [28,32].

To reveal the mechanisms of AAIL-enhanced oil recovery, the extraction process in Section 2.3 was simplified for simulation by Material Studio software as well. The molecular dynamics simulation process was conducted in two stages: (i) the adsorption of asphalt molecules on carbonate rocks, and (ii) the desorption of these asphalt when AAILs and solvents are introduced into the systems. In present work, the carbonate rocks was replaced by a calcite crystal with the lattice parameters of a = 29.94 Å, b = 29.94 Å and c = 15.64 Å and crystal plane angles of α = 90°, β = 90°, and γ = 120°. Then the calcite crystal was optimized with energy minimization and converted into the 3D periodic cells by building vacuum slabs. Actually, in asphalt, the asphaltene are the most polar fractions which possess the strongest affinity to the solid surfaces compared with other fractions. Herein, to be simplified, during adsorption simulation task, 5 asphaltene molecules were used to represent asphalt and were placed on top of calcite crystal in a simulation box of 29.94 Å × 29.94 Å × 80 Å. After the adsorption simulation, 20 AAIL molecules and 80 solvent molecules were placed on top of asphaltene which adhered on calcite crystal in the simulation box. The forcite module with COMPASS force field in NVT ensemble was used in all of the simulations. The temperature was set to 30 °C using a Berendsen thermostat. For the calculations of van der Waals and electrostatic interactions, the cut-off distance was fixed at 12.5 Å. The simulation time was 500 ps and the time step was 1.0 fs. The trajectories were collected in an interval of 1.0 ps for further analysis.

## 3. Results and Discussion

### 3.1. Characterizations of AAILs

^1^H NMR spectra of the synthesized AAILs were recorded and shown in Appendix A. Take ChPro for example. The ^1^H NMR spectra of Pro and ChPro were shown in Figure 3. Compared with those in Figure 3a, the new chemical shifts at 3.07 (labelled as 5′), 3.39 (labelled as 6′), and 3.92 ppm (labelled as 7′), were assigned to the N–(CH_3_)_3_, N–CH_2_– and –CH_2_– groups, respectively. Furthermore, proton signals which were assigned to the pyrrolidine ring, are found to shift downfield from δ = 1.96, 2.30, 3.30~3.37, and 4.09 ppm to δ = 1.62, 2.00, 2.67, 2.94, and 3.39 ppm (labelled as 1′, 2′, 3′, and 4′) respectively. These changes in proton signals are ascribed to the reaction between carboxylic acid and hydroxide ion. This chemical shift proves the generation of salt. According to the ratio of integration value of proton signals, the purity of synthesized IL is high.

The FT-IR spectra of ChCl and the synthesized AAILs were shown in Figure 4a–f. The characteristic peaks of ChCl (Figure 4a) appear at 3215 cm^−1^ (–OH stretching) and 630 cm^−1^ (C–Cl stretching). The characteristic peaks of the synthesized AAILs (Figure 4f) are located at 3350 cm^−1^ (–NH_2_ stretching and –OH stretching), 1571 cm^−1^ (–COO^−^ stretching) and 1491 cm^−1^ (–COO^−^ twisting). The absence of the signals at 630 cm^−1^ on the spectra of synthesized AAILs (Figure 4b–f) suggested that there is no unreacted ChCl residual in synthesized AAILs, and the synthesized AAILs possess high purity.

Based on the DSC curves (shown in Appendix A), the glass transition temperatures are observed in the range from −83.17 °C to −49.71 °C for all the ILs (shown in Table 2). All the AAILs are liquids at room temperature. The TGA curves of AAILs are shown in Appendix A. The decomposition temperatures are calculated from the intersection of the baseline and the tangent line in the TGA curves (shown in Table 2). The decomposition temperatures range from 138 °C to 185 °C, which means these AAIL are thermally stable enough for IL-assisted solvent extraction test.

### 3.2. Application in Carbonate Asphalt Rocks Separation

The AAIL-toluene mixture systems are used to extract heavy hydrocarbons from the Indonesia carbonate asphalt rocks. Results in Figure 5 show all of the one-stage asphalt recoveries of AAILs, ranked from 87.85% to 91.26%, are significantly higher than that of traditional toluene extraction (77.79%), especially ChHis (Figure 5a) up to 91.26%. It suggests that AAILs with conjugated structure (ChHis and ChPhe (Figure 5a,c)) perform better than other AAILs. Furthermore, it is also found that the viscosity of AAILs (shown in Table 2) exerts a slight influence on asphalt recovery. However, the interfacial tension of AAIL-toluene interface (shown in Table 3) plays an important role in influencing the asphalt recovery. AAILs with conjugated structure (e.g., ChPhe, 6.43 mN/m) have both lower interfacial tension of AAIL-toluene interface and higher recovery than AAILs without conjugated structure (e.g., ChGly, 19.08 mN/m). In addition to the asphalt recovery, the solid entrainment in the product from AAIL-toluene mixture systems is also highly improved, as shown in Table 4. 

Figure 6a–f show the FT-IR measurement results of the extracted asphalt after IL-assisted extraction. The characteristic peaks of asphalt (Figure 6f) appear at 2923, 2856 cm^−1^ (C–H stretching), and 1456, 1372 cm^−1^ (C–H scissoring). The absence of the signals at 1571 cm^−1^ (–COO stretching) and 1491 cm^−1^ (–COO^−^ twisting), which are assigned to AAILs (Figure 4), on the spectra of extracted asphalt (Figure 6a–e) suggested that the AAIL dissolution in the asphalt is negligible. On the other hand, the solubility of asphalt and solvent in AAILs were calculated using COSMOthem software, shown in Table 5. The simulation results showed that all the SARA fractions and toluene are insoluble in AAILs, which are consistent with the experimental tests.

### 3.3. Computational Simulation for AAIL-Assisted Solvent Extraction

In previous work, we found that ILs possess strong interactive forces with the mineral surface [11,26]. To investigate the role of AAILs in enhancing oil recovery, the interaction energy calculations of AAILs-carbonate interfaces were conducted by molecular dynamics simulation. The more negative the interaction energy is, the stronger the adsorption is. To be simplified, the asphaltene (one fraction of the asphalt) was taken as an example. The calculation results (shown in Table 6) indicate the value of interaction energy between calcite crystal and AAILs (e.g., ChHis, −2019.61 kcal/mol) is smaller than that between calcite crystal and asphaltene (−91.51 kcal/mol). It suggests that the electrostatic forces play a dominant role in adsorption process. It also means that AAILs (especially, ChHis) have stronger interactive forces with the calcite surface than asphaltene. The results demonstrate that ILs facilitate the asphaltene (oil fraction) detaching from the carbonate surface.

The surface free energies of SARA molecules at the corresponding SARA fraction bulk phase-AAIL interface (Figure 7a) or solvent-AAIL interface (Figure 7b) were simulated by the COSMOtherm software. A smaller surface free energy means it is easier for asphalt components to build a new surface at the interface. For each AAIL, the surface free energies of SARA fraction molecules at toluene-AAIL interfaces are smaller than those at the SARA-AAIL interface, suggesting that the transfer of SARA fractions from SARA phase to toluene through the IL phase would happen spontaneously. In other words, it is hard for SARA fractions to transfer from toluene phase to asphalt phase through IL phase as “non-return valve”. These results suggest that the presence of AAILs could facilitate the SARA fractions transferring from the asphalt phase to toluene phase, leading to a higher dissolution of asphalt in toluene and a higher recovery.

According to the results above, it proves that the AAILs play a “carrier” role in the IL-assisted solvent extraction from carbonate asphalt rocks, carrying the asphalt fractions from the calcite surface to the solvent phase for dissolution.

Similar results were obtained from the molecular dynamics simulation. The spontaneous desorption process of asphaltene from a modelled carbonate asphalt rock surface immersed in AAIL-toluene mixture is displayed in Figure 8. Initially, the asphalt molecules clustered tightly due to their fused-ring conjugate plane structure, and adhered to calcite surface with aliphatic long alkyls. When AAILs were added into the system, the cation and anion of AAILs rapidly squeezed into the gaps between asphalt clusters and replaced them on the surface due to the strong electrostatic forces of the cation and anion of AAILs. In this stage, the asphalt clusters was carried away from surface while the toluene diffused tardily towards asphalt clusters. Finally, toluene molecules adhered to asphalt clusters, which means asphalts were dissolved by solvent. Furthermore, the forcite dynamics energies results also reveal the non-band energy of simulation system plays a dominant role which was ascribed to the strong electrostatic forces of AAILs.

### 3.4. The Selection of AAIL for Enhancing Oil Recovery

The designability is the greatest charm of IL. In this study, five kinds of AAILs were designed via introducing functional group into amino acetic acid anion (ChGly), such as phenyl group (ChPhe), imidazole group (ChHis), hydroxy group (ChSer), and pyrrolidine group (ChPro), to modify the physical and chemical properties of IL. The properties and behavior of AAILs, such as viscosity (shown in Table 2), surface/interfacial tension (shown in Table 3), solubility of asphalt and toluene in AAIL (shown in Table 5), interaction energy calculation of AAIL-carbonate interfaces (shown in Table 6), and surface free energies of asphalt composition at asphalt-AAIL interface (shown in Figure 7), were characterized and calculated. According to the results above, ChHis possesses the smallest value of interaction energy of AAILs-carbonate interface, the smallest value of surface free energy of asphalt composition at asphalt-AAIL interface, the smallest solubility of asphalt and toluene and a relatively low interfacial tension of AAIL-toluene interface. ChHis is suggested as the optimal one among the series of AAILs for separating asphalt from the carbonate rocks, which is in accordance with the results of experimental tests (shown in Figure 5). It is indicated that the introduction of functional group conjugated structure (i.e., imidazole ring) into IL improves the interactions between asphalt and IL and between the solvent and IL because of the structural similarity. 

## 4. Conclusions

A series of AAILs were synthesized economically using cheap raw materials via a one-pot synthesis, achieving high purity. These AAILs were found to perform well in separating the extra-heavy oil from carbonate asphalt rocks, resulting in better oil quality with fewer solid entrained compared with solvent extraction. This is because AAILs play a “carrier” role in the IL-assisted solvent extraction from carbonate asphalt rocks, carrying the asphalt fractions from the calcite surface to the solvent phase for dissolution. Moreover, the simulated calculation results select choline histidine, as the optimal one among the series of AAILs, for enhancing oil recovery, since the introduction of functional group with conjugated structure enhances the interaction between IL and asphalt. The simulation results are in accordance with the results of the real test.

## Figures and Tables

**Figure 1 nanomaterials-09-00504-f001:**
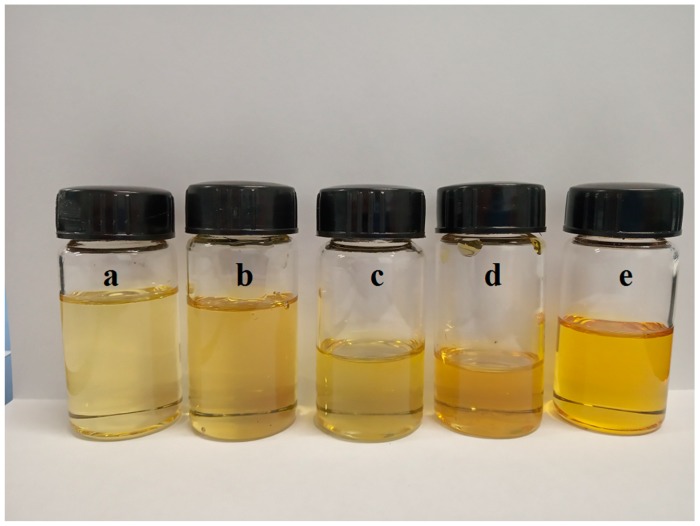
Photographs of samples under ambient conditions: (**a**) choline glycine (ChGly), (**b**) choline histidine (ChHis), (**c**) choline serine (ChSer), (**d**) choline proline (ChPro), and (**e**) choline phenylalanine (ChPhe).

**Figure 2 nanomaterials-09-00504-f002:**
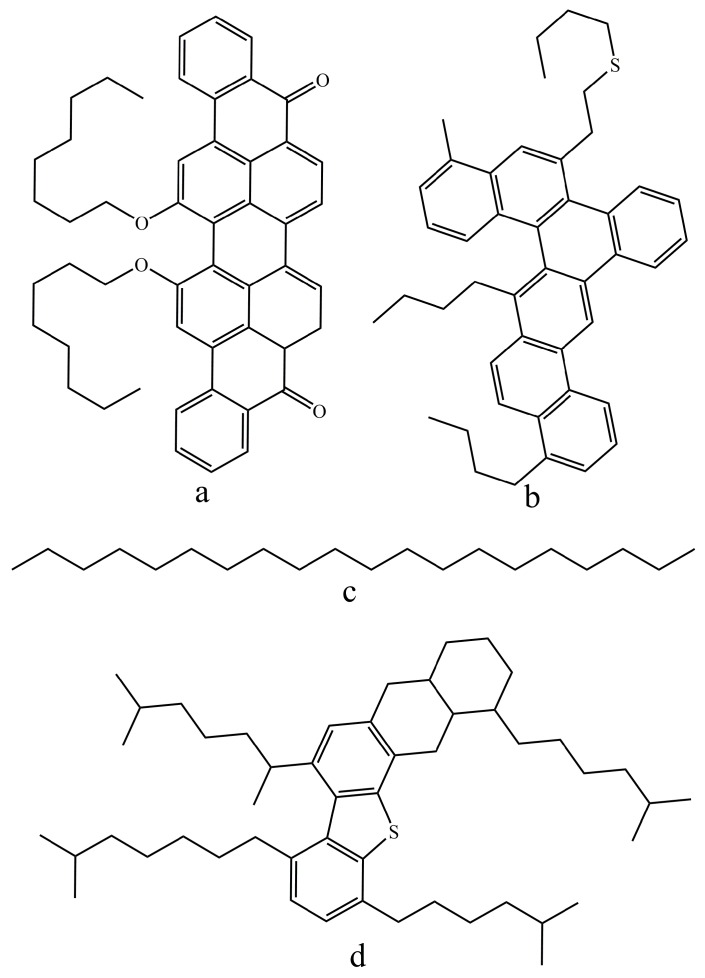
Molecular structures of bitumen component (SARA) fractions: (**a**) asphaltenes (C_50_H_48_O_4_), (**b**) aromatics (C_46_H_50_S), (**c**) saturates (C_20_H_42_), and (**d**) resins (C_50_H_80_S).

**Figure 3 nanomaterials-09-00504-f003:**
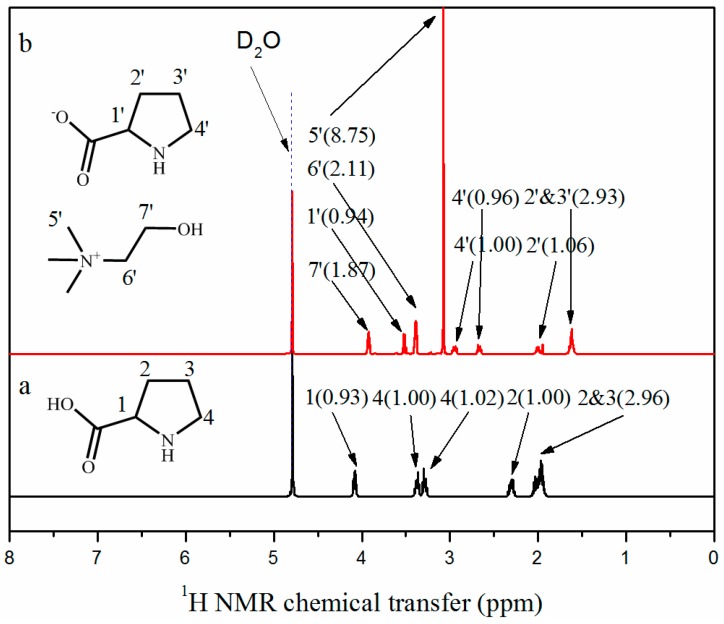
^1^H nuclear magnetic resonance spectra of (**a**) Pro and (**b**) ChPro. The solvent was D_2_O as indicated by peaks near 4.79 ppm.

**Figure 4 nanomaterials-09-00504-f004:**
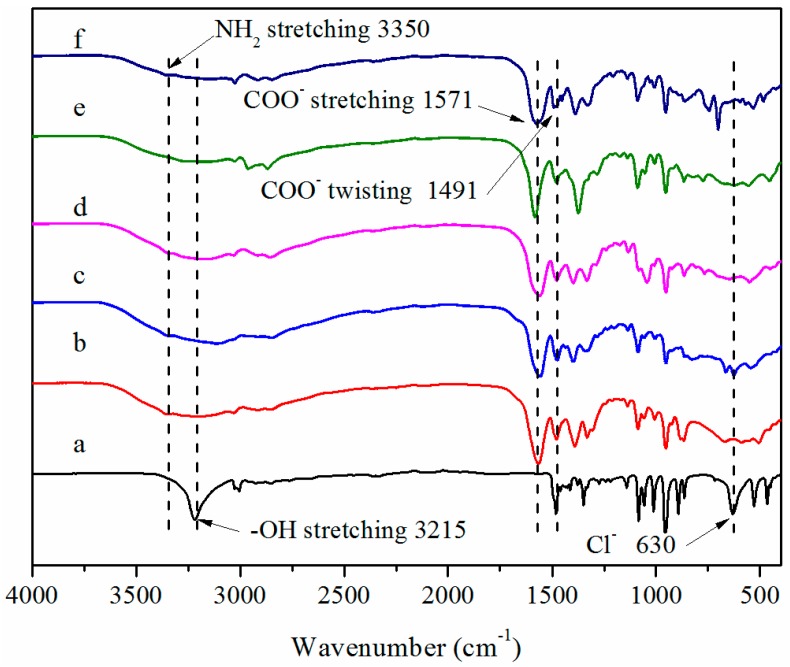
Fourier transform infrared spectroscopy spectra of (a) ChCl, (b) ChGly,(c) ChHis, (d) ChSer, (e) ChPro, and (f) ChPhe.

**Figure 5 nanomaterials-09-00504-f005:**
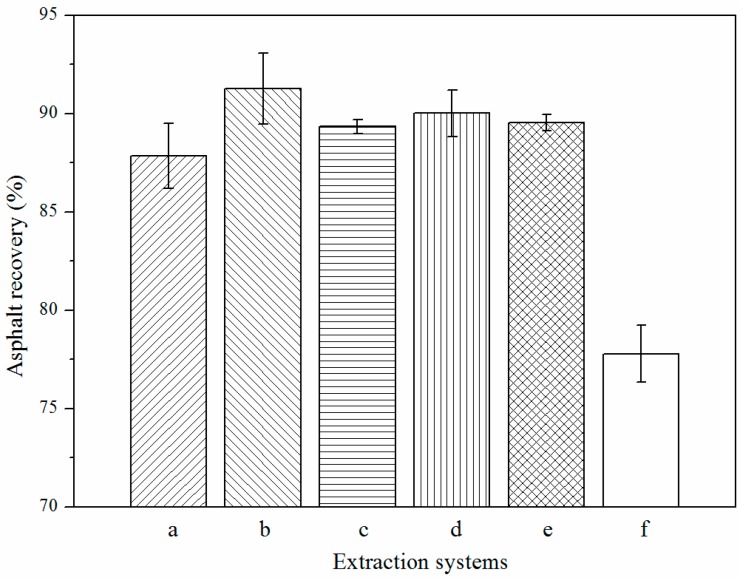
Asphalt recovery by solvent extraction by: (a) ChGly-toluene mixture, (b) ChHis-toluene mixture, (c) ChSer-toluene mixture, (d) ChPro-toluene mixture, (e) ChPhe-toluene mixture, and (f) toluene.

**Figure 6 nanomaterials-09-00504-f006:**
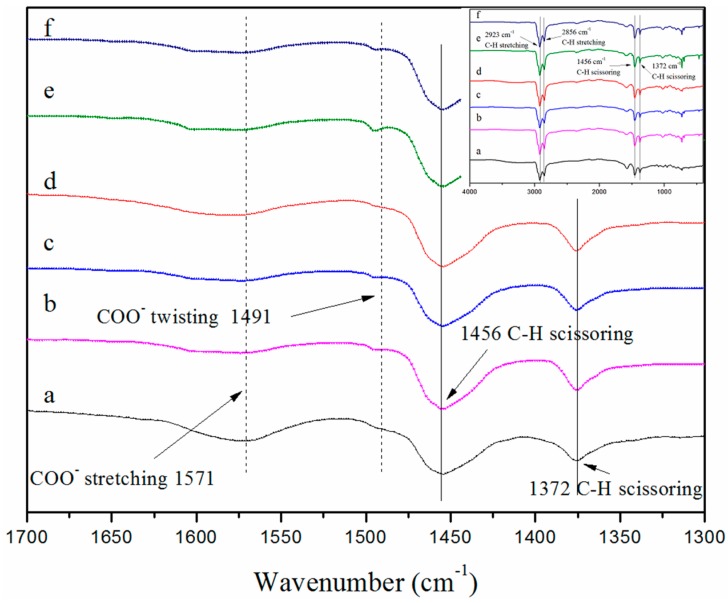
FT-IR spectra of: (a) asphalt from ChGly-assisted toluene extraction, (b) asphalt from ChHis-assisted toluene extraction, (c) asphalt from ChSer-assisted toluene extraction, (d) asphalt from ChPro-assisted toluene extraction, (e) asphalt from ChPhe-assisted toluene extraction, and (f) asphalt from pure toluene extraction.

**Figure 7 nanomaterials-09-00504-f007:**
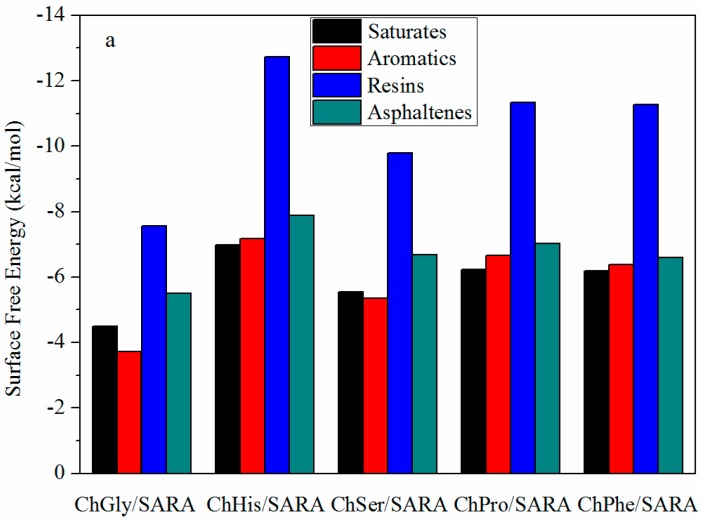
Surface free energies of different SARA molecules at specific interfaces with the corresponding asphalt extraction recovery: (**a**) IL-SARA fraction interfaces and (**b**) toluene-IL interfaces.

**Figure 8 nanomaterials-09-00504-f008:**
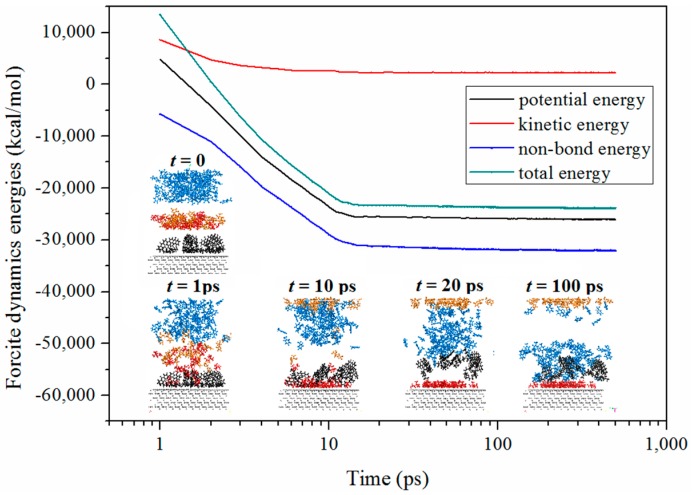
Consecutive snapshot of spontaneous desorption of asphaltene from a modelled carbonate asphalt rocks surface immersed in AAIL-toluene mixture and forcite dynamics energies of simulation system. Blue = toluene; red = cation of AAIL; orange = anion of AAIL; black = asphaltene and grey = calcite surface.

**Table 1 nanomaterials-09-00504-t001:** Physicochemical Properties of the Amino Acids used in this Study.

Name	Structure	ρ ^a^ (g/cm^−3^)	pI ^b^	T_m_ ^c^ (°C)
Glycine (Gly)	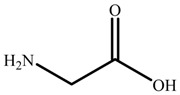	1.595	5.97	240
Histidine(His)	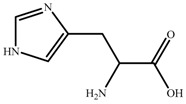	1.309	7.59	282
Serine (Ser)	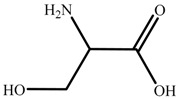	1.530	5.68	240
Proline (Pro)	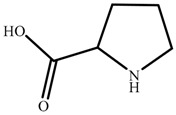	1.350	6.30	228
Phenylalanine (Phe)	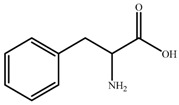	1.290	5.48	283

^a^ Density; ^b^ Isoelectric point; ^c^ Melting point.

**Table 2 nanomaterials-09-00504-t002:** Properties of the amino acid ionic liquids (AAILs).

	T_g_ (°C )	T_d_ (°C)	Viscosity (mPa·s) ^a^	Density (g/cm^−3^)	Surfase Tension (mN/m) ^b^
ChGly	−83.17	138.57	32.08	1.140	61.81
ChHis	−51.15	177.75	570.3	1.129	59.22
ChSer	−62.53	184.56	352.5	1.156	57.66
ChPro	−69.78	165.08	272.8	1.111	55.86
ChPhe	−49.71	163.35	443.8	1.121	56.01

^a^ Viscosity at 30 °C; ^b^ Surfase tension at 30 °C.

**Table 3 nanomaterials-09-00504-t003:** Surface tension of AAIL and interfacial tension of AAIL-toluene interface.

	ChGly	ChHis	ChSer	ChPro	ChPhe
Surface tension of AAIL (mN/m)	61.81	59.22	57.66	55.86	56.01
Interfacial tension of AAIL-toluene interface (mN/m)	19.08	8.48	15.92	9.04	6.43

**Table 4 nanomaterials-09-00504-t004:** Solids entrained in asphalt-toluene solution in different extraction systems.

	ChGly-Toluene	ChHis-Toluene	ChSer-Toluene	ChPro-Toluene	ChPhe-Toluene	Toluene
Solids entrained (mg/mL)	3.00 ± 0.38	1.41 ± 0.29	2.25 ± 0.31	2.63 ± 0.49	1.13 ± 0.29	13.17 ± 0.42

**Table 5 nanomaterials-09-00504-t005:** Calculated solubility (x, mol/mol) of SARA fractions from asphalt and solvents in AAILs.

	Saturates	Aromatics	Resins	Asphaltenes	Toluene
ChGly	7.01 × 10^−5^	2.80 × 10^−4^	2.96 × 10^−7^	2.26 × 10^−3^	7.16 × 10^−2^
ChHis	7.75 × 10^−7^	4.80 × 10^−7^	3.55 × 10^−11^	3.13 × 10^−6^	3.86 × 10^−2^
ChSer	1.04 × 10^−5^	1.29 × 10^−5^	5.93 × 10^−9^	8.59 × 10^−5^	5.06 × 10^−2^
ChPro	3.03 × 10^−6^	1.27 × 10^−6^	4.03 × 10^−10^	6.03 × 10^−6^	4.78 × 10^−2^
ChPhe	3.25 × 10^−6^	2.03 × 10^−6^	4.53 × 10^−10^	1.03 × 10^−5^	6.31 × 10^−2^

**Table 6 nanomaterials-09-00504-t006:** Interaction energy calculation of AAILs-carbonate interfaces.

	E_total_(kcal/mol)	E_adsorbate_(kcal/mol)	E_surface_(kcal/mol)	E_interaction_(kcal/mol)
E_elctrostatic_	E_van der Walls_	E_elctrostatic_	E_van der Walls_	E_elctrostatic_	E_van der Walls_	
ChGly-calcite	23460.41	275.89	25114.15	−1929.63
−15706.20	39170.22	270.76	5.21	−14027.70	39144.45	
ChHis-calcite	23463.15	271.39	25211.37	−2019.61
−15710.42	39177.45	263.04	8.513	−13930.65	39144.45	
ChSer-calcite	23454.89	292.02	25112.27	−1949.39
−15729.79	39188.45	283.86	8.26	−14029.58	39144.45	
ChPro-calcite	23469.95	266.68	25147.58	−1944.31
−15725.47	39199.21	261.38	5.42	−13994.33	39144.45	
ChPhe-calcite	23500.08	281.42	25226.52	−2007.85
−15673.39	39177.41	270.49	11.12	−13915.53	39144.45	
Asphaltene-calcite	25406.75	39.18	25459.07	−91.51
−13718.42	39131.62	37.11	3.32	−13683.35	39144.45

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
