# Peer review of "Novel Synthesis of Choline-Based Amino Acid Ionic Liquids and Their Applications for Separating Asphalt from Carbonate Rocks"

_nanomaterials, 2019, doi:10.3390/nano9040504_

Round 1

Reviewer 1 Report

Authors have chosen an area of research which fits well to the wide readership of the journal. The manuscript is very well written and usage of English is of acceptable standards. The manuscript content is technically sound and the discussion includes detailed observations and inferences. I appreciate the efforts taken to carry out this research work.

I recommend the manuscript for publication.

Author Response

Response to Reviewer 1 Comments

Point 1:Authors have chosen an area of research which fits well to the wide readership of the journal. The manuscript is very well written and usage of English is of acceptable standards. The manuscript content is technically sound and the discussion includes detailed observations and inferences. I appreciate the efforts taken to carry out this research work.

I recommend the manuscript for publication.

Response 1: We appreciate for your approval, and take this opportunity to thank you again.

Reviewer 2 Report

This paper is estimating the potential of a series of choline-based amino acid ionic liquids as an assistive solvents for extracting asphalt from carbonate rocks. This paper is interesting because the idea of this paper is a new approach to extracting bitumen from carbonate rocks. However, the mechanisms for the bitumen recovery by the mixtures is not shown clearly in the current manuscript. Therefore, the paper could be improved in several ways:

- Table 1

The structure of Proline and Phenylalanine should be also shown in this table.

- p.3, L.85

Why was the mole ratio of choline chloride, potassium hydroxide and amino acid decided as 1:1:1.05? Please show the clear grounds for the ratio.

- p.4, L.107

Why was the weight ratio of AAIL and toluene decided as 1:2? Please show the clear grounds for the ratio.

- p.4, L.105

What shape of rock sample was added into the liquid mixture in this study? Powder, granular, or bulk? Although the weight of ore sample was shown in this paper, the size of ore sample such as particle size, grain size, or bulk size should be also shown.

- p.4, L.108

The elemental composition of the ore (carbonate rock) sample which was used in this study should be shown in this paper.

- Table 2

Are the AAILs the newtonian fluids or non-newtonian fluids? If they are non-newtonian fluids, please show the share rate at which you measured the viscosity of them

- p.7, L.210

This paper is insisting that the interfacial tension of AAIL-toluene interface plays an important role in influencing the asphalt recovery. However, I don't think so because there is no clear relevance between the trend of the interfacial tension and that of the bitumen recovery. For example, the mixture ChHis brought the highest recovery although the interfacial tension of that mixture was the second highest. Similarly, the mixture ChPro brought the second highest recovery although the interfacial tension of that mixture was the fourth highest. Please reconsider the discussion on those results.

- p.4, L.117

Please show the definition of the solids entrainment in the asphalt by showing an equation.

- Figure 5

The mixtures are shown in the order of ChGly, ChHis, ChSer, ChPro, and ChPhe in all the Tables. On the other hand, Figure 1, Figure 5, Figure 6, and Figure 7 show the results of those mixtures in different order. It is easier for readers to understand the Figures which show the results of the mixtures in the same order as shown in the Tables. Please rearrange the order of the mixtures on those Figures in the same order.

- Table 3

Why is the trend of surface tension and interfacial tension different? For example, the surface tension of AAIL containing ChPro (50.91) is lower than that of AAIL containing ChPhe (56.01) whereas the interfacial tension of toluene-AAIL containing ChPro (19.18) was higher than that of toluene-AAIL containing ChPhe (6.43). Please add a description explaining the reasons.

- Table 5

The unit of solubility is not shown clearly. Please show the unit of solubility.

- Figure 7

The color of the bars showing the Asphaltenes in Figure 7 is different between a (moss green) and b (pink). The color of bars showing Asphaltenes should be unified between both figures.

Author Response

Response to Reviewer 2 Comments

Point 1: Table 1. The structure of Proline and Phenylalanine should be also shown in this table.

Response 1: Thank you for your reminder. Done as suggested.

Point 2: p.3, L.85. Why was the mole ratio of choline chloride, potassium hydroxide and amino acid decided as 1:1:1.05? Please show the clear grounds for the ratio.

Response 2: Thanks the reviewer for this comment. Theoretically, the mole ratio of reactants should be 1:1:1, according to Eq. 1. In this process, An extra 5% amino acid was used to make sure choline chloride and potassium hydroxide could react completely, since the unreacted choline chloride is hard to separate from the amino acid ionic liquid. However, the excess amino acids are easy to remove because they are not solvable in acetonitrile. On the other hand, wasting further more amino acid would influence the purification of AAIL, the 5% excess is enough.

Point 3: p.4, L.107. Why was the weight ratio of AAIL and toluene decided as 1:2? Please show the clear grounds for the ratio.

Response 3: Thank you for pointing this out. We had tested different weight ratios of carbonate asphalt rock, AAIL and toluene and found 1:1:2 was the optimum one. We found the more AAIL couldn’t offer higher asphalt recovery, but the less dosage of AAIL would influence asphalt recovery. The influence of dosage of toluene on asphalt recovery is the similar. Taking the material cost and energy consumption into consideration, we chose the weight ratio of carbonate asphalt rock, AAIL and toluene as 1:1:2.

Point 4: p.4, L.105. What shape of rock sample was added into the liquid mixture in this study? Powder, granular, or bulk? Although the weight of ore sample was shown in this paper, the size of ore sample such as particle size, grain size, or bulk size should be also shown.

Response 4: Thank you for your instructive suggestions. It has been done as suggested, shown in the revised manuscript, in Lines 79~81, Page 4.

Point 5: p.4, L.108. The elemental composition of the ore (carbonate rock) sample which was used in this study should be shown in this paper.

Response 5: Thanks for your valuable suggestion. Done as suggested in the revised manuscript, in Lines 79~81, Page 4.

Point 6: Table 2. Are the AAILs the newtonian fluids or non-newtonian fluids? If they are non-newtonian fluids, please show the share rate at which you measured the viscosity of them

Response 6: Thank you for pointing this out. As suggested, we measured rheological behaviors of the five AAILs in this study, and found All of them are the newtonian fluids, The viscosity of AAILs is independent on the shear rate. Similar study had been done otherwhere, and got the same results, according to Figure 1S from Reference 22: Ionic liquids from renewable biomaterials: synthesis, characterization and application in the pretreatment of biomass.(Liu, Q.P., et al., Green Chemistry, 2012. 14(2): p. 304-307).

Point 7: p.7, L.210. This paper is insisting that the interfacial tension of AAIL-toluene interface plays an important role in influencing the asphalt recovery. However, I don't think so because there is no clear relevance between the trend of the interfacial tension and that of the bitumen recovery. For example, the mixture ChHis brought the highest recovery although the interfacial tension of that mixture was the second highest. Similarly, the mixture ChPro brought the second highest recovery although the interfacial tension of that mixture was the fourth highest. Please reconsider the discussion on those results.

Response 7: Thank you for your valuable and thoughtful comments. To verify the conclusion,we measured the surface tension of AAIL and interfacial tension of AAIL-toluene interface again and found some results presented in table 3 were wrong. We are very sorry for this. In the revised manuscript, the corresponding results were corrected. According to the Table 3 and Figure 5, it is obviously found that the ionic liquid with low interfacial tension of AAIL-toluene interface show higher the asphalt recovery, i.e. ChHis and ChPhe, indicating that the interfacial tension of AAIL-toluene interface plays an important role in influencing the asphalt recovery. However, the extraction process was a complicated multiphase separation, the interfacial tension of AAIL-toluene interface is importance but not the only factor on the asphalt recovery. We also found that ChPhe showed lowest interfacial tension of AAIL-toluene interface, but ChPhe exhibited the third highest the asphalt recovery. That is to say, the asphalt recovery is also affected by other factor, i.e. surface free energy of AAIL- asphalt interface, which is consisted with the simulation results in Figure 7.

Point 8: p.4, L.117. Please show the definition of the solids entrainment in the asphalt by showing an equation.

Response 8: Thank you for pointing this out. As suggested, we have defined the solids entrainment in the revised manuscript, shown in Lines 122~124, Page 4.

Point 9: Figure 5. The mixtures are shown in the order of ChGly, ChHis, ChSer, ChPro, and ChPhe in all the Tables. On the other hand, Figure 1, Figure 5, Figure 6, and Figure 7 show the results of those mixtures in different order. It is easier for readers to understand the Figures which show the results of the mixtures in the same order as shown in the Tables. Please rearrange the order of the mixtures on those Figures in the same order.

Response 9: Thank you for your valuable advice. As suggested, to make the reading and understanding much easier, Figure 1, Figure 5, Figure 6, and Figure 7 have been renewed and the order of AAILs has been rearranged, shown in the revised manuscript.

Point 10: Table 3. Why is the trend of surface tension and interfacial tension different? For example, the surface tension of AAIL containing ChPro (50.91) is lower than that of AAIL containing ChPhe (56.01) whereas the interfacial tension of toluene-AAIL containing ChPro (19.18) was higher than that of toluene-AAIL containing ChPhe (6.43). Please add a description explaining the reasons.

Response 10: Thank you for your instructive suggestions. According to the previous literature, Surface tension, interfacial tension and contact angles of ionic liquids (Rossen Sedev, Current Opinion in Colloid & Interface Science 2011(16) 310–316) the interfacial tension of AAIL-toluene interface was influenced by the structure of anion of AAIL. For example, ChPhe possesses benzyl group so that the interfacial tension of ChPhe -toluene interface is the lowest.

Point 11: Table 5 The unit of solubility is not shown clearly. Please show the unit of solubility.

Response 11: Great thanks for the reviewer again. As suggested, to make the reading and understanding much easier, the caption of Table 5 has been checked and rephrased shown in the revised manuscripts. The new caption is given as followed: Table 5. Calculated Solubility (x, mol/mol) of SARA Fractions from asphalt and solvents in AAILs.

Point 12: Figure 7. The color of the bars showing the Asphaltenes in Figure 7 is different between a (moss green) and b (pink). The color of bars showing Asphaltenes should be unified between both figures.

Response 12: Great thanks for the reviewer again. As suggested, to make the reading and understanding much easier, Figure 7 has been renewed, shown in the revised manuscripts.

Round 2

Reviewer 2 Report

Thank you for carefully addressing each of my comments. The quality of the manuscript has clearly further improved from the previous version and is now suitable for publication.